# Application of Multi-Dimensional Intelligent Visual Quantitative Assessment System to Evaluate Hand Function Rehabilitation in Stroke Patients

**DOI:** 10.3390/brainsci12121698

**Published:** 2022-12-10

**Authors:** Yuying Du, Yu Shi, Hongmei Ma, Dong Li, Ting Su, Ou Zhabayier Meidege, Baolan Wang, Xiaofeng Lu

**Affiliations:** 1Department of Rehabilitation Medicine, The First Affiliated Hospital of Xinjiang Medical University, Urumqi 830054, China; 2School of Communication and Information Engineering, Shanghai University, Shanghai 200444, China

**Keywords:** assessment, rehabilitation, stroke, upper extremity, hand

## Abstract

Background: Hand dysfunction is one of the main symptoms of stroke patients, but there is still a lack of accurate hand function assessment systems. This study focused on the application of the multi-dimensional intelligent visual quantitative assessment system (MDIVQAS) in the rehabilitation assessment of hand function in stroke patients and evaluate hand function rehabilitation in stroke patients. Methods: Eighty-two patients with stroke and unilateral hand dysfunction were evaluated by MDIVQAS. Cronbach’s Alpha coefficient was used to assess the internal consistency of MDIVQAS; the F-test is used to assess the differences in MDIVQAS for multiple repeated measures. Spearman’s analysis was used to identify correlations of MDIVQAS with other assessment systems. *t*-tests were used to identify differences in outcomes assessed with MDIVQAS in patients before and after treatment. *p* < 0.05 were considered significant. Results: (1) Cronbach’s Alpha coefficient of MDIVQAS in evaluating hand’s function > 0.9. (2) There was no significant difference between the other repeated measurements, except for thumb rotation in MDIVQAS. (3) MDIVQAS had a significant correlation with other assessment systems (r > 0.5, *p* < 0.01). (4) There were significant differences in the evaluation of hand function in patients before and after treatment using MDIVQAS. Conclusion: The MDIVQAS system has good reliability and validity in the evaluation of stroke hand function, and it can also better evaluate the treatment effect.

## 1. Introduction

Stroke is a major noncommunicable disease that harms the health of people all over the world. As of 2019, stroke is the second leading cause of death worldwide and the third leading cause of death caused by disability. From 1990 to 2019, the disability rate caused by stroke increased by 32.0% [1], and 70% to 90% of patients after stroke have upper limb dysfunction [2,3], 60–70% of them with hand dysfunction [1,4]. As one of the most important organs of the human body, the hand is involved in many daily activities, and its function accounts for 90% of the upper limb function. Hand dysfunction seriously affects patients’ daily life and social participation ability [5]. Rehabilitation therapy is an important method to restore hand function, and accurate treatment is based on an accurate assessment. At present, the commonly used methods for assessing hand function after stroke are mainly qualitative or semi-quantitative, which mainly include the protractor measurement of Active Range of Motion (AROM), FMA-UE, FMA-W/H, Brunnstrom, ARAT and so on. The above evaluation methods have been widely used in clinical practice for a long time, and their reliability and validity are also widely recognized. However, the actual application is easily influenced by personal experience [6,7,8,9]. Therefore, there is great interest in developing an automated system to achieve intelligent, objective and quantitative assessment of hand function rehabilitation after stroke. In the last two decades, significant advances have been made in human motion measurement and analysis, providing the technical basis for automated assessment of upper limb and hand function [10]. In recent years, modern methods for quantitative hand function assessment include the 3D motion capture system, Kinect technology, rehabilitation robots and micro sensors, etc. However, the above instruments are still lacking in large sample studies, and some instruments are too bulky and have not been widely used in clinical practice [11,12,13].

In clinical rehabilitation work, intelligent, quantitative, accurate and efficient hand function assessment tools are needed to provide a better choice for clinical practice. Therefore, this study uses a newly developed intelligent assessment method, application of multi-dimensional intelligent visual quantitative evaluation system (MDIVQAS) to objectively and quantitatively assess the function of the affected hand after stroke.

MDIVAQAS is the core technology of hand motion calculation jointly developed by Huashan Hospital affiliated with Fudan University and Shanghai University with completely independent intellectual property rights. Through the optical smart capture and the integration of computer vision technology, complete animation action standard hand more guidance, the contralateral national health model and subject to lateral hand evaluation process, belongs to the field initiative, and it can complete the quantitative assessment of hand function. The purpose of this study was to verify the reliability of MDIVQAS, and to observe the evaluation effect of the system in clinical application compared with traditional evaluation methods.

## 2. Materials and Methods

### 2.1. General Information

Using the pwr package and WebPower package in R, it is assumed that there is a large effect size, statistical testing power 1-β = 0.8 and significance level a = 0.05, and the minimum sample size required for each stage is calculated. From November 2019 to October 2021, 88 stroke patients who met the inclusion criteria and signed informed consent in the Department of Rehabilitation Medicine, the First Affiliated Hospital of Xinjiang Medical University were selected. Inclusion criteria: ① The diagnosis was in line with the diagnostic criteria for stroke (including cerebral infarction and cerebral hemorrhage) formulated by the Cerebrovascular Department of the Neurology Branch of the Chinese Medical Association in 2019 [14]. ② One-hand dysfunction, and in the hand of Brunnstrom recovery stages (BRS-H) II and above. ③ The National Institute of Health Stroke Scale (NIHSS) score ≤ 4, the nerve damage was mild, the condition was relatively stable. ④ Modified Ashworth Scale (MAS) score < level 2. ⑤ Sitting balance ≥ 2 level and can remain seated for 60 min. ⑥ no serious defects in communication, memory or understanding, and the ability to follow the instructions of the assessor; ⑦ Willing to cooperate with the completion of the subject assessment, signed informed consent.

Exclusion criteria: ① Congenital or prior to the onset of the malformation of the affected hand due to other reasons, limited joint activity, etc. ② Motor dysfunction of both hands; ③ the condition is unstable. ④ Poor compliance, patients or family members refused to participate. This study has received ethics approval of hospital (20200624-11). After recruitment, Figure 1 represents the number of samples in each stage.

### 2.2. Methods

The rehabilitation physician makes a clear diagnosis and refers the patient to a systematically trained rehabilitation therapist, who explains the purpose of the assessment to the patient, informs the assessment content, demonstrates the specific actions before the assessment, obtains the relevant information of the patient and informs the relevant procedures and precautions before the test, so that the patient can fully understand and cooperate.

Before rehabilitation treatment, the enrolled patients were selected to complete the reliability and validity verification. Fifty-seven males completed the comparison and evaluation of the healthy hand modeling and patient hand of the 10 movements of MDIVQAS, among which twenty-four patients completed three repeated measurements by the same examiner with an interval of no more than 24 h. After the completion of MDIVQAS, all enrolled patients underwent the propiometric measurement of AROM, FMA (FMA-UE, FMA-W/H), Brunnstrom (upper limb, hand) and ARAT assessment, once each, within 24 years in sequence. The above assessment was repeated 2 weeks after rehabilitation.

Enrolled patients were allowed a short rest between each method of assessment. All patients were required to complete the assessment at the same test site and under the same test conditions. The total assessment time for each patient was approximately 60 min.

The specific assessment methods were as follows:(1)MDIVQAS: Based on the pathological motor characteristics of hemiplegic hand and a set of post-stroke hand function rehabilitation evaluation actions corresponding to the Brunnstrom scale, Fugl-Meyer Rating Scale and range of motion measurement, it is a computer-aided technology-based assessment tool. Using the comprehensive quantitative evaluation method of healthy hand modeling and comparison evaluation of the affected hand, the 3D spatial position and motion vector information of various joints of the phalanx, metacarpal and wrist were acquired in real time with the help of video equipment, and then various motion parameters of the hand joint were analyzed as the system parameters of the hand function evaluation standard. In order to prevent the ambiguity and subjectivity in the guidance process of the standard movement demonstration, At the bottom left of the screen, there is a 3D animation of the action being evaluated to achieve a consistent demonstration of standard hand movements. The assessment items included three parts as forearm, wrist and hand, with a total of 10 movements, including ulnar wrist deviation, wrist dorsiextension, five fingers adduction and abduction, forearm pronation, forearm supination, spherical grip, cylindrical grip, thumb flexion and extension, thumb abduction and thumb rotation.

When the patient was seated, the patient information was first input, such as number, name, age, brief medical history, assessment results of common scales, etc. The pathological information of the patient included healthy hand, affected hand, stroke type, stroke brain area, etc. The hospital information includes the name, address and contact information of the hospital. Then, the movements to be evaluated were selected. The patient placed the healthy hand and the affected hand in the evaluation device at the same time, and the same visual and optical acquisition devices were configured for the healthy hand and the affected hand, respectively. In the first step of evaluation, the patient’s healthy hand was guided by the standard animated hand to complete the extraction of the motion characteristics and node parameters of the patient’s healthy hand and the modeling of the healthy hand model. The 3D animated hand part was captured by Maya software. The standard hand video obtained the foreground hand through the background learning algorithm, and the spatial motion trajectory was tracked by the hand particle filter. In the second step of assessment, the affected hand completed the extraction of motion characteristics and joint parameters in the affected hand working area. In the third step of evaluation, the multi-dimensional hand motion parameters of the patient’s affected hand to be evaluated were comprehensively analyzed, and the joint motion of the affected hand, the percentage of joint motion of the affected hand in the healthy hand and the evaluation time were automatically calculated. In the third step of evaluation, the multi-dimensional hand movement parameters of the patient’s hand to be evaluated are comprehensively analyzed, and the joint range of motion of the affected hand, the percentage of joint mobility of the affected hand to the healthy hand and the time of evaluation are automatically calculated. Evaluate twice, and the system will automatically select the better angle result for saving. MDIVQAS flow block diagram, see Figure 2.

(2)Measuring AROM with protractor: A universal protractor was used to measure the forearm pronation, forearm supination, ulnar deviation, wrist dorsiextension and the angle between the fingers of the five fingers [9].(3)FMA-UE [15,16]: It mainly includes movement, speed, coordination and reflex activities, with a total of 66 points, and each item is scored on a 3-level scale: that is, 0 points, unable to perform; 1 point, partially implemented; 2 points, fully implemented. Among them FMA-W/H is a part of the FMA rating scale, which evaluates the wrist and hand. There are 12 items in total, each item is 0~2 points, full score is 24 points. The higher the score, the better the motor function of the upper limb is indicated.(4)Brunnstrom Scale [7,17]: upper limb and hand parts; each is divided into stage I–VI, and the higher the level, the better the motor function. Stage I: no exercise; Stage II: slight flexion; Stage III: flexion but not extension; Stage IV: the thumb can be pinched and loosened, and the fingers can be extended semi-randomly in a small area; Stage V: can do spherical or cylindrical grip, and can be free to extend the whole finger, but the range of size is not equal; Stage VI: full range extension of various grips, but with less speed and accuracy than the healthy side.(5)ARAT [18,19]: Consisting of 4 subscales (grasp, grip, pinch and gross motion), which mainly evaluates the ability of the affected hand to handle objects of different sizes, weights and shapes. ARAT requires a standardized assessment toolbox, consisting of 19 items with a full score of 57, and each item is scored in a 4-point order (0: unable to complete any part of the task within 60 s, 1: complete part of the task within 60 s, 2: The task is completed, but the difficulty is very high or the time is too long (5~60 s), 3 points: the normal completion within 5 s). Each of ARAT’s subscales is arranged in a hierarchical order, testing the most difficult items first, then the easiest and then increasing the items in turn. The higher the score, the better the feature.

### 2.3. MDIVQAS

#### 2.3.1. Overall Design Scheme of MDIVQAS

MDIVQAS was a fully independent intellectual property technology jointly developed by Huashan Hospital affiliated with Fudan University and Shanghai University. By collecting video signals and conducting computer vision analysis, the automatic detection and dynamic tracking based on hand position are completed. Then, combined with the intelligent voice prompt module, the real-time hand movements and spatial positions of patients in the process of healthy hand modeling and affected hand assessment are dynamically detected. Through optical intelligent motion capture equipment, real-time acquisition of three-dimensional space information and motion vector data of each joint point of palm, finger and wrist. Aiming at the current hand function rehabilitation training qualitative assessment Brunnstrom scale and other corresponding sets of post-stroke hand function rehabilitation assessment actions such as forearm pronation or supination, wrist radial deviation, wrist ulnar deviation and wrist dorsiextension, thumb adduction and abduction, four fingers (except the thumb) adduction and abduction and other movements, combined with function of this platform hand movements intelligent analysis algorithm software module, feature data dimension reduction and mode matching, analysis of hand joint movement parameters, as the hand function recovery system evaluation parameters of quantitative evaluation criteria. At the same time, through multi-point network connection, a remote management server was set up to achieve quantitative, accurate, standardized and consistent quality management of the evaluation process through data spot check and video monitoring. Using the new idea of unilateral healthy hand to guide the affected hand, the animation standard hand guidance of multiple actions, the modeling of the healthy hand on the healthy side and the assessment of the affected hand on the affected side were completed. A set of standardized and comprehensive quantitative assessment process and implementation methods based on standard hand, patient’s healthy hand and patient’s affected hand were designed to scientifically and quantitatively solve the problem of phased quantitative assessment of hand function rehabilitation after stroke.

#### 2.3.2. Hardware Platform of MDIVQAS

The platform was mainly composed of the hardware of the operating platform and the computing software of hand function evaluation, which is in line with the characteristics of patient ergonomics and motion. In this study, the industrial design process was used to carry out a complete industrial product-level design through the form frame structure mold opening, the integrated optimization of video and optical motion capture equipment, and the integrated configuration of touch control. The data collected by each hospital can be uploaded to the cloud platform, so as to obtain big data for quantitative evaluation of hand function of different populations for effective data classification and analysis [20]. The evaluation tool has declared a total of 5 patent achievements, of which 3 are invention patents, 1 is design patent and 1 is utility model patents. The project team further developed the identity authentication function based on face recognition, which has been tested in the cloud system, enabling patients to use “face brushing”, greatly reducing the complexity of login and other procedures. The core hardware structure frame of this platform was shown in Figure 3.

The touch display was responsible for the input and output of interactive information. Computer was responsible for data processing, logic control, intelligent analysis algorithm and information storage functions. The left and right workspaces were the detection areas of the platform, and patients can complete the functions of modeling the healthy hand and assessing the affected hand. Dual-channel video acquisition equipment could complete video signal acquisition.

A four-way optical motion capture device (developed by Shanghai University, Shanghai, China) was used to obtain real-time 3D spatial data and multiple motion vector information of each joint point of the patient’s finger, palm and wrist. The external structure frame was the overall scaffold of the platform. The light source provided a good illumination environment for the working area, reduced noise and improved detection accuracy. A physical photo of the hardware platform prototype of MDIVQAS was shown as Figure 4.

### 2.4. Statistical Analysis

SPSS17.0 statistical software was used for statistical processing. Measurement data with normal distribution and homogeneity of variance were expressed as *x* ± s. Univariate *t*-test was used for comparison between the two groups, and Pearson coefficient was used for correlation. The data that did not meet the above conditions were described by the median (interquartile range), the comparison between the two groups was performed by Wilcoxon’s Kolmogorov–Smirnov Z (K–S) test and the correlation analysis was by Spearman’s test. Cronbach’s Alpha coefficient and repeated measures were used to analyze the internal consistency of the assessment system. The pwr package in R was used to analyze the required sample size in the study. *p* < 0.05 was considered statistically significant.

## 3. Results

General data from November 2019 to October 2021: 88 stroke patients who met the inclusion criteria and signed informed consent in the Department of Rehabilitation Medicine, the First Affiliated Hospital of Xinjiang Medical University, were selected, of which six patients were missed (two missed the proposed evaluation time, three failed to cooperate and one dropped out midway). A total of 82 patients, including 57 males (69.5%) and 25 females (30.5%), completed the evaluation of the multi-dimensional intelligent visual quantitative assessment system, with an average age of (54.29 ± 13.12) years.

### 3.1. Reliability of MDIVQAS

The consistency test preset large effect size f = 0.4 [21], statistical testing power 1-β = 0.8, significance level a = 0.05 and at least 18 subjects were required. Considering the possibility of sample loss in the process of clinical research, the sample size was appropriately increased by 10% [22], and the results showed that at least 20 subjects were needed. This sample size was used to guide the content consistency test of this study.

The 24 patients enrolled in the group completed 10 movements using MDIVQAS, and each movement was repeated three times by the same examiner. Cronbach’s alpha coefficient method was used to analyze the internal consistency of the assessment system. All the 10 actions were greater than 0.9, indicating that the internal homogeneity reliability of MDIVQAS was excellent, and the internal consistency was good. See Table 1 for details.

Each patient enrolled was completed by MDIVQAS, and each action was evaluated three times within 24 h. Statistical analysis of the internal consistency of the assessment system by repeated measures showed that there was a statistical difference in the consistency test of thumb rotation (*p* < 0.05). However, no statistical difference was found in the repeated measurement of other movements, and there was consistency. See Figure 5 and Appendix A for details.

### 3.2. Validity of MDIVQAS

#### 3.2.1. Correlation between MDIVQAS, FMA-W/H, Brunnstrom and ARAT Assessment

The correlation had a statistically large effect size f^2^ = 0.35 [21], statistical test power 1-β = 0.8 and significance level a = 0.05. At least 30 subjects are needed, considering the possibility of sample loss in the process of clinical study, and the sample size was appropriately increased by 10% on this basis [22], the results showing that at least 33 subjects were needed. This sample size was used to guide the correlation test of this study.

Among the subjects who met the inclusion criteria, Brunnstrom (upper limb) at stage IV or above and Brunnstrom (hand) at stage II or above completed the measurements of the percentage of affected side functions in the healthy side functions of wrist dorsiextension, wrist ulnar deviation, finger adduction and abduction, spherical grip, cylindrical grip, thumb flexion and extension and thumb rotation. Within 24 h, the same patient was given the Fugl-Meyer Assessment of Wrist and Hand (FMA-W/H), which is the wrist–hand assessment part of FMA-UE, Brunnstrom and ARAT scale. Statistical analysis was performed on the correlation between the percentage of the affected hand on the unaffected side measured by MDIVQAS and the various scales. Bivariate Correlations, Pearson’s Correlations and two-tailed tests were used for the statistical parameters. The results showed that MDIVQAS was strongly correlated with FMA-W/H, Brunnstrom (hand) and ARAT (r > 0.5, *p* < 0.01). See Figure 6 and Appendix A.

#### 3.2.2. Correlations MDIVQAS and Protractor Measurement

The patients who met the inclusion criteria used the multi-dimensional intelligent visual quantitative assessment system and the protractor to measure the AROM of joint for the movement of forearm pronation, forearm supination, wrist dorsiextension, wrist ulnar deviation and finger adduction and abduction 1 (angle between the thumb and index finger of the affected hand); finger adduction and abduction 2 (angle between the index finger and the middle finger of the affected hand); finger adduction and abduction 3 (angle between the middle finger and the ring finger of the affected hand) and finger adduction and abduction 4 (angle between the ring finger and the little finger of the affected hand) within 24 h, respectively. Statistical analysis was performed on the correlation of the AROM of joint of the same movement for the two methods. Bivariate Correlations, Spearman’s Correlations and two-tailed tests were used to analyze the correlations. The correlation coefficient (r) between MDIVQAS and the protractor measurement in the above actions were all >0.5, indicating a strong correlation. See Figure 7 and Appendix A.

### 3.3. Reactivity before and after Treatment

#### 3.3.1. Comparison of Differences of MDIVQAS, FMA-UE, FMA-W/H, Brunnstrom and ARAT before and after Treatment

The difference test between the two groups was presupposed to have a large effect size d = 0.8 [21], statistical testing power 1-β = 0.8 and significance level a = 0.05, and at least 26 subjects were required. Considering the possibility of sample loss in the process of clinical research, the sample size was appropriately increased by 10% [22]. The results showed that at least 29 participants were required.

The enrolled patients used FMA-UE, FMA-W/H, Brunnstrom and ARAT before and 2 weeks after rehabilitation treatment to explore the differences before and after treatment. FMA-UE, FMA-W/H, Brunnstrom and ARAT all met the normality test (*p* > 0.05) and were described as *x* ± s. Paired *t*-test was used to analyze the differences between the assessment methods before and after treatment. The results showed that the differences of the above five assessments before and after treatment were statistically significant (*p* < 0.05), suggesting that the results of hand function evaluation of patients after treatment were improved compared with those before treatment. As shown in Figure 8 and Appendix A.

The patients who met the inclusion criteria were measured with a protractor before and after rehabilitation treatment to measure the AROM in finger adduction and abduction. Nonparametric Tests: Two related samples were used to analyze the differences before and after treatment, and the results showed that there were statistically significant differences in adduction and abduction between the fingers before and after treatment (*p* < 0.05), as shown in Figure 9 and Appendix A.

The enrolled patients were evaluated by MDIVQA before rehabilitation treatment and 2 weeks after treatment to evaluate the percentage of the affected hand in the healthy hand. The data of wrist ulnar deviation, wrist dorsiextension, finger adduction and abduction, forearm pronation, forearm supination, cylindrical grip, spherical grip, thumb abduction, thumb flexion and extension and thumb rotation did not meet the normality test (*p* < 0.05) and were described by the median (interquartile range). Nonparametric Tests 2: Related samples was used to analyze the differences assessed by MDIVQA before and after rehabilitation treatment, and the results suggested that the above actions had statistical significance before and after treatment (*p* < 0.05). It is suggested that MDIVQA could sensitively assess changes in patients’ hand functions, as shown in Figure 10 and Appendix A.

#### 3.3.2. Comparison of the Difference between MDIVQAS and Protractor Measurement of AROM in the Increase of Joint Motion before and after Treatment

The reactivity of the two evaluation methods to the treatment effect was basically the same, and there was no statistical significance in the increase of the range of motion between the two evaluation methods before and after treatment (*p* > 0.05), indicating that the two evaluation methods had the same reactivity to the treatment effect. See Table 2.

## 4. Discussion

Hand function plays an important role in people’s daily life, affecting people’s working, eating, dressing, modifying and other activities. The improvement of hand and upper limb function will maximize the recovery of overall function and improve the quality of life of stroke patients. Effective rehabilitation requires objective, quantitative, effective and reliable rehabilitation assessment [23]. Photoelectric capture technology in intelligent evaluation tools is considered as the gold standard of human motion analysis [24]. MDIVQAS in this study is a newly developed intelligent evaluation method using optical intelligent capture technology. This system is a hand function assessment tool jointly developed by Huashan Hospital affiliated with Fudan University and Shanghai University. It uses optical intelligent motion capture equipment and computer vision technology to conduct hand modeling and hand evaluation and obtain three-dimensional spatial data and motion vector information of each point of fingers, palms and wrists. At present, it has been able to carry out specific intelligent analysis algorithm for the 10 movements of five fingers adduction and abduction, wrist ulnar deviation, wrist dorsiextension, spherical grip, cylindrical grip, thumb flexion and extension, thumb abduction, thumb rotation, forearm pronation and forearm supination, and the exercise parameters of the healthy hand angle value, the affected hand angle value and the affected hand/healthy hand ratio were analyzed. At present, the feasibility study and quantitative evaluation application of the equipment with small samples of normal volunteers have been carried out [25,26]. In the early stage, the research team tested the semi-reliability and duplicate reliability in terms of reliability, and the reliability coefficients are both >0.850, indicating that the system has good reliability, consistency and stability. The reliability of the 10 actions of MDIVQAS showed a statistically significant difference in the reliability of the evaluators (*p* < 0.01), indicating that the reliability of MDIVQAS retest was high, and good and stable results could be obtained by repeating the measurement in a short period of time. In terms of validity test, the content validity test of MDIVQAS in the early stage of our research team showed that all 10 movements were common hand dysfunction after stroke, the I-CVI of the entry level was 1 and each action had a high correlation with the total score (*p* < 0.01), suggesting that it had certain evaluation value. The structural validity test adopts the exploratory factor analysis, a total of one common factor is extracted and the cumulative variance contribution rate is >60%, according to the functional component of the action, indicating that the system has good structural validity and can well reflect the hand motor function, and the structural validity test results show that MDIVQAS is single-dimensional in terms of evaluation content, has strong pertinence and is suitable for quantitative evaluation of hand function. In terms of the convergence validity test, the AVE of the system > 0.500, indicating that it has good convergence validity [27].

The team expanded the sample size of the previous study and, at the same time, used MDIVQAS to test the intra-group consistency of 10 movements of the affected hand of stroke patients, and Cronbach’s alpha > 0.9, indicating good internal consistency. The repeatability of 10 actions was measured, and the results showed that the differences in the repeated measurement of nine actions were not statistically significant (all *p* > 0.05), indicating that the system had good repeatability. Only one of the movements (thumb rotation) had a statistically significant difference (F = 3.603, *p* = 0.045), indicating that the repeatability of this action needs to be explored. In conclusion, the above evidence shows that MDIVQAS has good confidence in the assessment of hand function after stroke. The consideration of the results of thumb rotation may be related to the fact that the current development of computer vision and pattern recognition algorithms has not reached the level of good recognition of any complex actions. When the dysfunctional hand of stroke patients is the hand, and the Brunnstrom stage ≥ IV on this side, the thumb rotation action is more flexible than the healthy hand (the hand that builds the model), the data exceed the modeling range and the data accuracy is reduced. According to the experience of the evaluator, when the hand function is relatively good or recovers to a certain extent, due to the flexibility of the hand, the completion of the action is better than the healthy side, and the accuracy of the data is reduced. It is recommended to debug and rectify the evaluation and measurement methods of the above actions.

At the same time, MDIVQAS was used in this study to test the calibration validity of AROM, FMA-UE, FMA-W/H, Brunnstrom and ARAT, and the results were all > 0.5 and *p* < 0.01, indicating that MDIVQAS was strongly correlated with the above four widely used clinical evaluation methods. It is suggested that MDIVQAS has good validity. Fugl-Meyer, Brunnstrom, ARAT and protractor measurements of AROM are the most commonly used classical methods in clinical evaluation of poststroke motor dysfunction [15,17,18]. Especially, FMA-W/H, FMA-UE and ARAT are the most commonly used scales to evaluate the efficacy of upper limb and hand motor function after stroke and are often used as the gold standard to test the validity of other scales [28]. This study confirmed that all the 10 movements evaluated by MDIVQAS had strong correlation with the above four evaluation methods, so MDIVQAS had good validity.

In this study, the evaluation methods of MDIVQAS and protractor measurement of AROM, Brunnstrom, FMA-W/H, FMA-UE and ARAT were put into clinical practice of stroke hand function rehabilitation, and the effects of hand function rehabilitation before and 2 weeks after treatment were evaluated. The results showed significant differences before and after treatment (*p* < 0.01~0.05), indicating that MDIVQAS could reflect the change of clinical treatment effects such as other classical methods. At the same time, the differences between MDIVQAS and a protractor to measure the increase in AROM before and after treatment were compared, and the results showed that there was no significant difference in the increase in the range of motion before and after treatment between the two evaluation methods (*p* > 0.05), indicating that the responsiveness of MDIVQAS to the treatment effect was consistent with that of the classical methods.

In clinical use, we have found that, first, the spherical grip, cylindrical grip and thumb rotation designed by MDIVQAS can make up for the shortcomings of traditional evaluation methods. Second, based on the automatic detection and dynamic tracking of hand position, combined with intelligent voice prompts and standard 3D animation guidance modules, the system facilitates the dynamic detection of real-time hand movements and spatial positions of patients in the process of healthy hand modeling and affected hand evaluation and realizes real-time dynamic evaluation of hand functions, which is simple to operate, convenient to use and dynamically adjusts and guides rehabilitation treatment plans. Third, the system can also set up a remote management server through a multi-point network connection, and the data collected by each hospital can be uploaded to the cloud platform so as to obtain the big data of the quantitative evaluation of the manual functions of different groups of people and can achieve quantitative, accurate, standardized and consistent evaluation process quality management through data spot checks, video monitoring and other methods. As more data accumulates, MDIVQAS can be optimized gradually. Fourth, the current intelligent quantitative motion evaluation methods for hand function are divided into wearable sensor schemes and noncontact vision schemes [10,29,30,31]. MDIVQAS is a noncontact evaluation scheme, which avoids the disadvantage that wearable sensor solutions restrict the freedom of movement of patients, especially for the evaluation of fine movements of the hand, which tends to produce large errors. Fifth, the system has certain limitations for subject selection, such as hand dysfunction, critical condition and people who cannot sit for a long time.

## 5. Conclusions

In conclusion, MDIVQAS has good reliability and validity in the evaluation of hand function in stroke, as well as good evaluation of the treatment effect. However, there are some shortcomings in the application that need to be further studied and improved.

## Figures and Tables

**Figure 1 brainsci-12-01698-f001:**
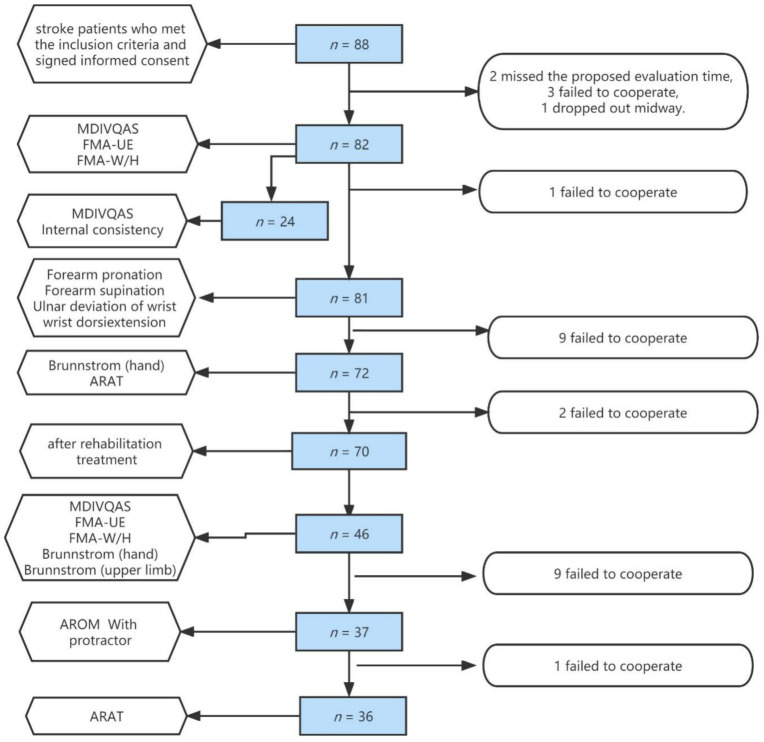
Flow chart of each stage in the study.

**Figure 2 brainsci-12-01698-f002:**
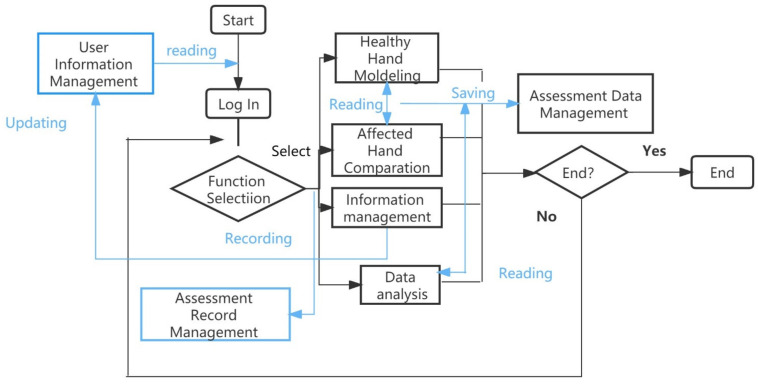
MDIVQAS workflow chart.

**Figure 3 brainsci-12-01698-f003:**
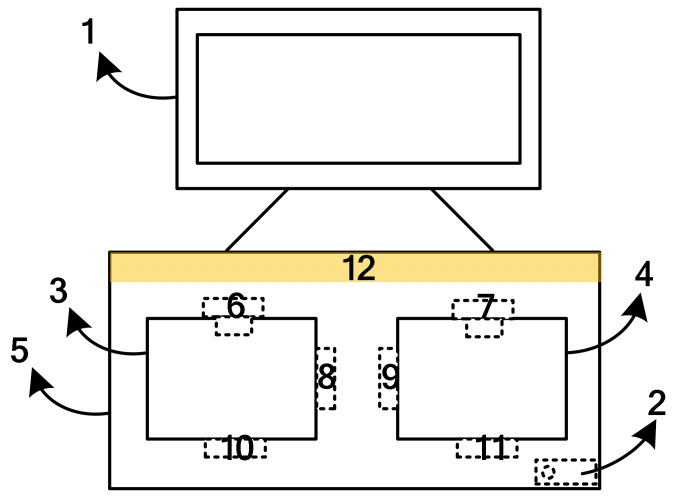
Block diagram of core hardware structure of hardware platform of multi-dimensional intelligent visual quantitative assessment system. 1—Touch display; 2—Computer; 3 and 4—Workspace; 5—External structure frame; 6 and 7—Video capture device; 8, 9, 10 and 11—Optical intelligent motion capture device and 12—Light source.

**Figure 4 brainsci-12-01698-f004:**
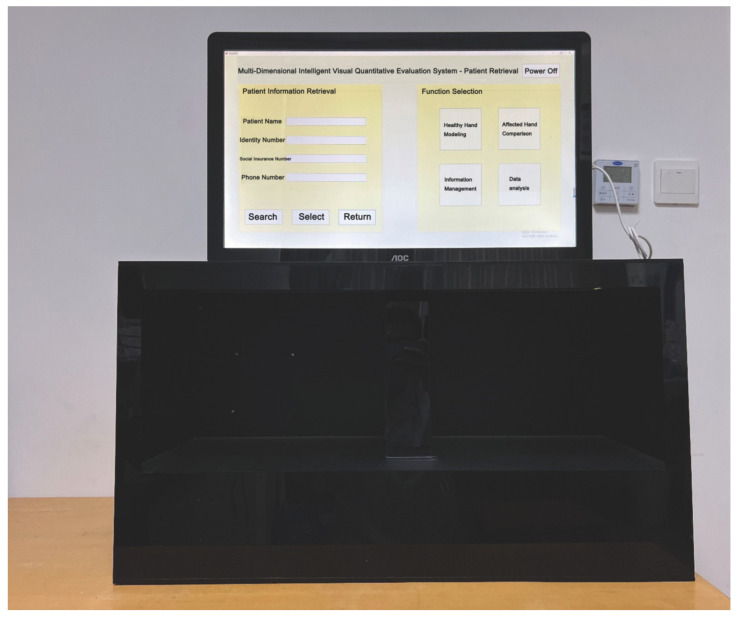
Hardware platform prototype of MDIVQAS.

**Figure 5 brainsci-12-01698-f005:**
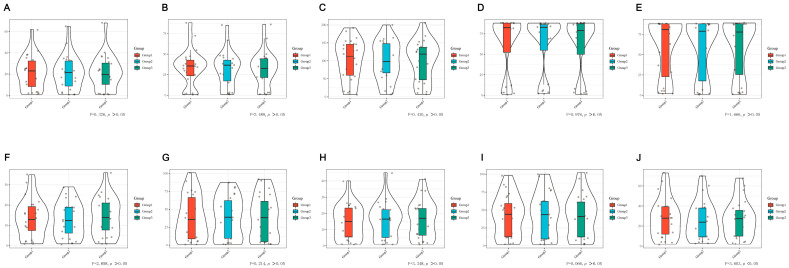
Consistency of the MDIVQAS for repeated measures to assess hand functions. (**A**) Wrist ulnar deviation; (**B**) Wrist dorsiextension; (**C**) Finger adduction and abduction; (**D**) Forearm pronation; (**E**) Forearm supination; (**F**) Cylindrical grip; (**G**) Spherical grip; (**H**) Thumb abduction; (**I**) Thumb flexion and extension; (**J**) Thumb rotation.

**Figure 6 brainsci-12-01698-f006:**
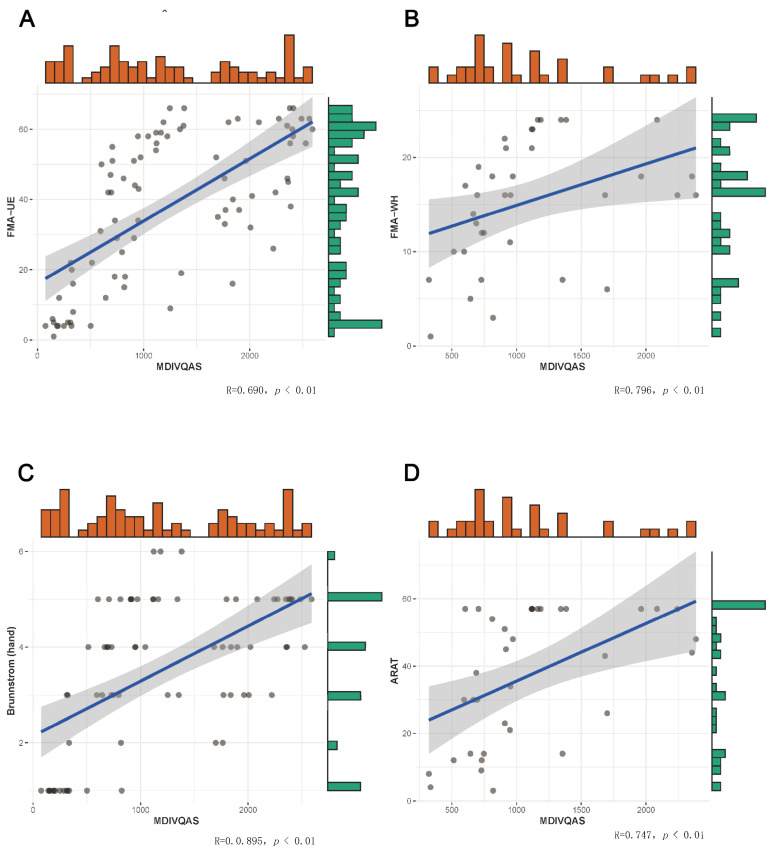
Correlation between MDIVQAS, FMA-W/H, Brunnstrom (hand) and ARAT. (**A**) Correlation between MDIVQAS and FMA-UE. (**B**) Correlation between MDIVQAS and FMA-W/H. (**C**) Correlation between MDIVQAS and Brunnstrom (hand). (**D**) Correlation between MDIVQAS and ARAT.

**Figure 7 brainsci-12-01698-f007:**
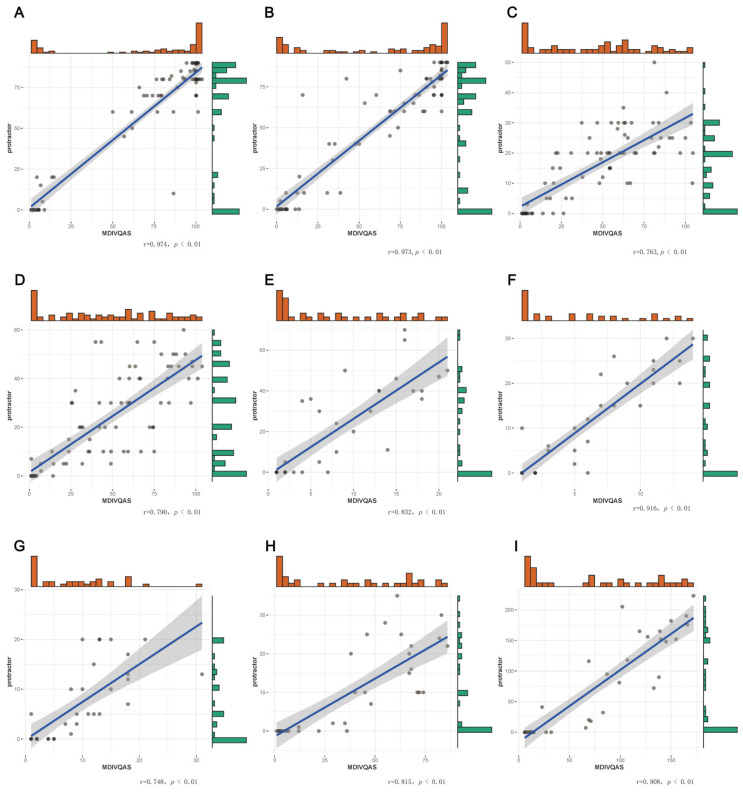
Correlation between MDIVQAS and protractor measurement. (**A**) Forearm pronation; (**B**) Forearm supination; (**C**) Ulnar deviation of wrist; (**D**) wrist dorsiextension; (**E**) Finger adduction and abduction1; (**F**) Finger adduction and abduction2; (**G**) Finger adduction and abduction3; (**H**) Finger adduction and abduction4; (**I**) Sum of finger adduction and abduction.

**Figure 8 brainsci-12-01698-f008:**
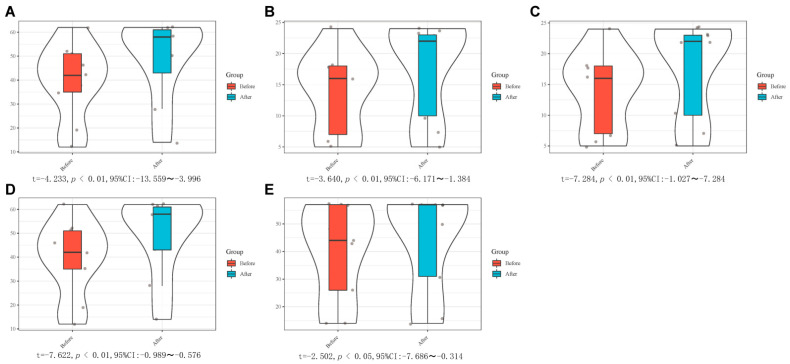
Comparison of differences of FMA-UE, FMA-W/H, Brunnstrom and ARAT before and after treatment. (**A**) FMA-UE; (**B**) FMA-W/H; (**C**) Brunnstrom (hand); (**D**) Brunnstrom (upper limb); (**E**) ARAT.

**Figure 9 brainsci-12-01698-f009:**
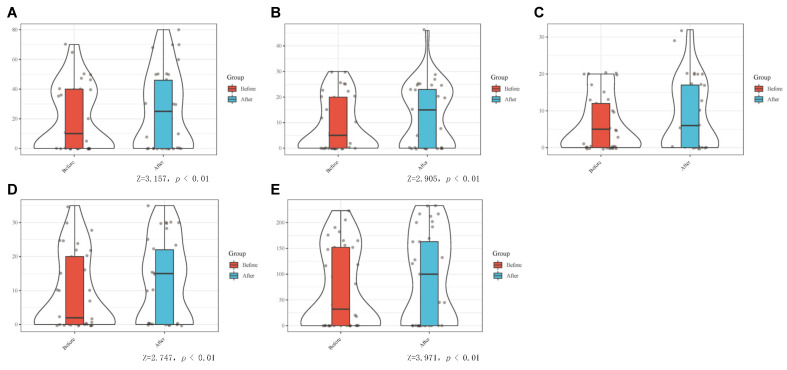
Comparison the differences of the AROM in finger adduction and abduction. With protractor before and after rehabilitation treatment. (**A**) Finger adduction and abduction 1; (**B**) Finger adduction and abduction 2; (**C**) Finger adduction and abduction 3; (**D**) Finger adduction and abduction 4; (**E**) Sum of finger adduction and abduction. Z represents the effect size of the two-sample K-S test (see Methods).

**Figure 10 brainsci-12-01698-f010:**
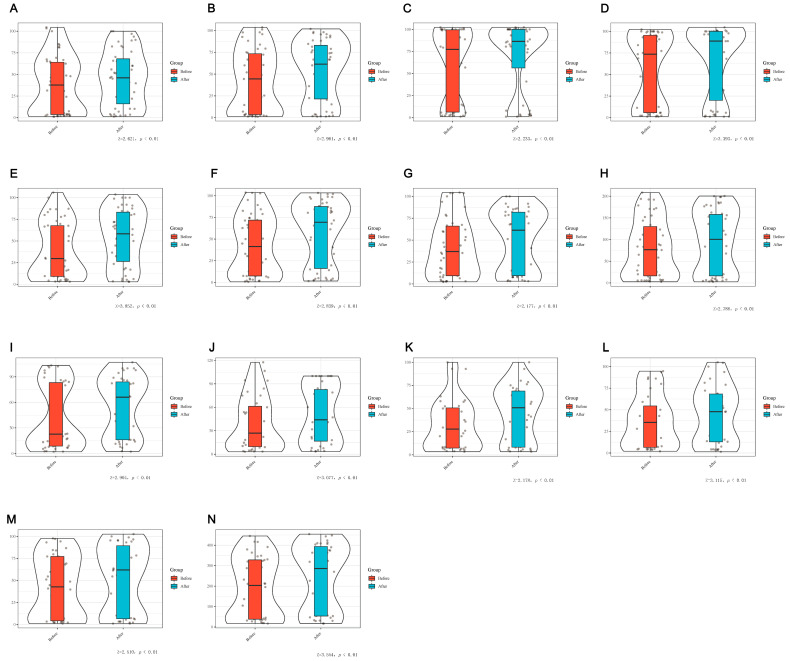
Comparison of differences before and after rehabilitation treatment using MDIVQAS. (**A**) Wrist ulnar deviation; (**B**) Wrist dorsiextension; (**C**) Forearm pronation; (**D**) Forearm supination; (**E**) Cylindrical grip; (**F**) Spherical grip; (**G**) Thumb abduction; (**H**) Thumb flexion and extension; (**I**) Thumb rotation; (**G**) Finger adduction and abduction 1; (**K**) Finger adduction and abduction 2; (**L**) Finger adduction and abduction 3; (**M**) Finger adduction and abduction 4; (**N**) Sum of finger adduction and abduction. Z represents the effect size of the two-sample K–S test (see Methods).

**Table 1 brainsci-12-01698-t001:** Reliability of MDIVQAS for assessing hand functions.

Movement (*n* = 24)	Cronbach’s Alpha	N of Items
Wrist ulnar deviation	0.989	3
Wrist dorsiextension	0.993	3
Finger adduction and abduction	0.987	3
Forearm pronation	0.998	3
Forearm supination	0.998	3
Cylindrical grip	0.981	3
Spherical grip	0.990	3
Thumb abduction	0.976	3
Thumb flexion and extension	0.989	3
Thumb rotation	0.994	3
Hand function 10 movements overall	0.989	30

**Table 2 brainsci-12-01698-t002:** Comparison of the MDIVQAS and protractor before and after treatment.

Item	n	P25	P50	P75	Z	P
Ulnar deviation increase A	37	0.0	1.0	5.0	−0.184 b	0.854
Ulnar deviation increase B	37	0.0	0.0	6.0		
Forearm pronation increase A	37	−1.5	1.0	11.5	−0.516 b	0.606
Forearm pronation increase B	37	0.0	3.0	10.0		
Forearm supination increase A	37	−0.5	2.0	20.0	−1.034 c	0.301
Forearm supination increase B	37	0.0	0.0	10.0		
Wrist dorsiextension increase A	37	−0.5	3.0	17.5	−0.403 c	0.687
Wrist dorsiextension increase B	37	0.0	3.0	15.0		
Increase in angle between the fingers 1A	37	−0.5	1.0	7.0	−1.267 b	0.205
Increase in angle between the fingers 1B	37	0.0	0.0	12.0		
Increase in angle between the fingers 2A	37	0.0	2.0	5.0	−0.502 b	0.616
Increase in angle between the fingers 2B	37	0.0	0.0	5.0		
Increase in angle between the fingers 3A	37	0.0	1.0	4.0	−0.868 b	0.386
Increase in angle between the fingers 3B	37	0.0	0.0	7.0		
Increase in angle between the fingers 4A	37	0.0	1.0	2.0	−1.783 b	0.075
Increase in angle between the fingers 4B	37	0.0	0.0	5.0		

Note: A: MDIVQAS B: protractor b: Based on the positive rank c: Based on the negative rank. Z represents the effect size of the two-sample K–S test (see methods).

## Data Availability

Data supporting the results of this study can be available by requesting the first author or corresponding author.

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
