# Peer review of "Application of Multi-Dimensional Intelligent Visual Quantitative Assessment System to Evaluate Hand Function Rehabilitation in Stroke Patients"

_brainsci, 2022, doi:10.3390/brainsci12121698_

Round 1

Reviewer 1 Report

In the introduction it would be good to explain in more detail why it is important to analyze the arm movements that are then evaluated in the study and what is their functional importance for the patients, and also to justify in the discussion the functional importance of the evaluation of these movements. 

In the methodology part, explain more graphically how the evaluations were carried out in the patients, explaining the number of sessions, the measurements and what was explained to the patients. Also describe better the type of activity that was asked to the patients to evaluate with the device if it was only movements that sequence took or if it was some activity that was described.

Author Response

Dear Editor,

Thank you for your email dated 22 November 2022 to inform us about the reviewers’ comments.

All the authors have seriously discussed all these comments. According to the

Reviewers’ comments, we have tried our best to modify our manuscript to meet the

requirements of your journal. In this revised version, changes to our manuscript within the

document were all highlighted in red colored text. Point-by-point responses are

listed below this letter. Your kind consideration is highly appreciated.

Yours sincerely,

Baolan Wang and Xiaofeng Lu

brainsci-2050099

Application of multi-dimensional intelligent visual quantitative assessment system to evaluate hand function rehabilitation in stroke patients

Response to Reviewer 1 Comments

Point 1: In the introduction it would be good to explain in more detail why it is important to analyze the arm movements that are then evaluated in the study and what is their functional importance for the patients, and also to justify in the discussion the functional importance of the evaluation of these movements. 

Response 1: Thank you for the reviewer’s advice, we have changed the introduction part list below, also in the manuscript’s lines 1-67, and hope it meets the criteria.

“Stroke is a major non-communicable disease that harms the health of people all over the world. As of 2019, stroke is the second leading cause of death worldwide and the third leading cause of death caused by disability. From 1990 to 2019, the disability rate caused by stroke increased by 32.0%[1] .70% to 90% of patients after stroke have upper limb dysfunction [2, 3] and 60-70% of them are hand dysfunction [1][4] . As one of the most important organs of the human body, hand is involved in many daily activities, and its function accounts for 90% of the upper limb function. Hand dysfunction seriously affects patients' daily life and social participation ability[5]. Rehabilitation therapy is an important method to restore hand function, and accurate treatment is based on accurate assessment. At present, the commonly used methods for assessing hand function after stroke are mainly qualitative or semi-quantitative, which mainly include: Protractor measurement of Active Range of Motion (AROM), FMA-UE, FMA-W/H, Brunnstrom, ARAT and so on. The above evaluation methods have been widely used in clinical practice for a long time, and their reliability and validity are also widely recognized. However, the actual application is easily influenced by personal experience [6-9]. Therefore, there is great interest in developing an automated system to achieve intelligent, objective and quantitative assessment of hand function rehabilitation after stroke. In the last two decades, significant advances have been made in human motion measurement and analysis, providing the technical basis for automated assessment of upper limb and hand function [10]. In recent years, modern methods for quantitative hand function assessment include 3D motion capture system, Kinect technology, rehabilitation robots and micro sensors, etc. However, the above instruments are still lacking in large sample studies, and some instruments are too bulky and have not been widely used in clinical practice[11-13].

In clinical rehabilitation work, intelligent, quantitative, accurate and efficient hand function assessment tools are needed to provide a better choice for clinical practice. Therefore, this study uses a newly developed intelligent assessment method, the ap-plication of a multi-dimensional intelligent visual quantitative evaluation system (MDIVQAS) to objectively and quantitatively assess the function of the affected hand after stroke.”

Point 2: In the methodology part, explain more graphically how the evaluations were carried out in the patients, explaining the number of sessions, the measurements and what was explained to the patients. Also describe better the type of activity that was asked to the patients to evaluate with the device if it was only movements that sequence took or if it was some activity that was described.

Response 2: We totally agree with the reviewer’s advice which helps make the manuscript be much clear. We have rewritten the methods part list below, also in the manuscript’s lines 95-194.

“2.2. Methods

The rehabilitation physician makes a clear diagnosis and refers the patient to a systematically trained rehabilitation therapist, who explains the purpose of the assessment to the patient, informs the assessment content, demonstrates the specific actions before the assessment, obtains the relevant information of the patient and in-forms the relevant procedures and precautions before the test, so that the patient can fully understand and cooperate.

Before rehabilitation treatment, the enrolled patients were selected to complete the reliability and validity verification. 82 patients completed the comparison and evaluation of the healthy hand modeling and patient hand of the 10 movements of MDIVQAS, among which 24 patients completed three repeated measurements by the same examiner with an interval of no more than 24 hours. After the completion of MDIVQAS, all enrolled patients underwent propiometric measurement of AROM, FMA (FMA-UE, FMA-W/H), Brunnstrom (upper limb, hand), and ARAT assessment, once each, within 24 years in sequence. The above assessment was repeated 2 weeks after rehabilitation.

Enrolled patients were allowed a short rest between each method of assessment. All patients were required to complete the assessment at the same test site and under the same test conditions. The total assessment time for each patient was approximately 60 minutes.

The specific assessment methods as follows:

①MDIVQAS: Based on the pathological motor characteristics of hemiplegic hand and a set of post-stroke hand function rehabilitation evaluation actions corresponding to Brunnstrom scale, Fugl-Meyer Rating Scale and range of motion measurement, it is a computer-aided technology-based assessment tool. Using the comprehensive quantitative evaluation method of healthy hand modeling and comparison evaluation of the affected hand, the 3D spatial position and motion vector information of various joints of the phalanx, metacarpal and wrist were acquired in real time with the help of video equipment, and then various motion parameters of the hand joint were analyzed as the system parameters of the hand function evaluation standard. In order to prevent the ambiguity and subjectivity in the guidance process of the standard movement demonstration, At the bottom left of the screen, there is a 3D animation of the action being evaluated to achieve a consistent demonstration of standard hand movements. The assessment items included three parts as forearm, wrist and hand, with a total of 10 movements, including ulnar wrist deviation, wrist dorsiextension, five fingers adduction and abduction, forearm pronation, forearm supination, spherical grip, cylindrical grip, thumb flexion and extension, thumb abduction and thumb rotation.

When the patient was seated, the patient information was first input, such as number, name, age, brief medical history, assessment results of common scales, etc. The pathological information of the patient included healthy hand, affected hand, stroke type, stroke brain area, etc. The hospital information includes the name, address, and contact information of the hospital. Then, the movements to be evaluated were selected. The patient placed the healthy hand and the affected hand in the evaluation device at the same time, and the same visual and optical acquisition devices were con-figured for the healthy hand and the affected hand respectively. In the first step of evaluation, the patient's healthy hand was guided by the standard animated hand to complete the extraction of the motion characteristics and node parameters of the patient's healthy hand and the modeling of the healthy hand model. The 3D animated hand part was captured by Maya software. The standard hand video obtained the foreground hand through the background learning algorithm, and the spatial motion trajectory was tracked by the hand particle filter. In the second step of assessment, the affected hand completed the extraction of motion characteristics and joint parameters in the affected hand working area. In the third step of evaluation, the multi-dimensional hand motion parameters of the patient's affected hand to be evaluated were comprehensively analyzed, and the joint motion of the affected hand, the percentage of joint motion of the affected hand in the healthy hand, and the evaluation time were automatically calculated. In the third step of evaluation, the multi-dimensional hand movement parameters of the patient's hand to be evaluated are comprehensively analyzed, and the joint range of motion of the affected hand, the percentage of joint mobility of the affected hand to the healthy hand, and the time of evaluation are automatically calculated. Evaluate twice, and the system will automatically select the better angle result for saving. MDIVQAS flow block diagram, see Figure 2.

Figure 2. workflow diagram

② Measuring AROM with protractor: A universal protractor was used to measure the forearm pronation, forearm supination, ulnar deviation, wrist dorsiextension and the angle between the fingers of the five fingers[9].

③FMA-UE [15, 16]: It mainly includes movement, speed, coordination and reflex activities, with a total of 66 points, and each item is scored on a 3-level scale, that is, 0 points, unable to perform; 1 point, partially implemented; 2 points, fully implemented. Among them FMA-W/H is a part of the FMA rating scale, which evaluates the wrist and hand. There are 12 items in total, each item is 0~2 points, full score is 24 points. The higher the score, the better the motor function of the upper limb is indicated.

④Brunnstrom Scale [7, 17]: upper limb and hand parts, each part is divided into stage I-VI, the higher the level, the better the motor function. Stage 1: no exercise; Stage Ⅱ: slight flexion; Stage Ⅲ: Flexion but not extension; Stage Ⅳ: The thumb can be pinched and loosened, and the fingers can be extended semi-randomly in a small ar-ea;Ⅴ: Can do spherical or cylindrical grip, can be free to extend the whole finger, but the range of size is not equal; Stage Ⅵ: Full range extension of various grips, but with less speed and accuracy than the healthy side.

⑤ARAT [18, 19]: Consisting of 4 subscales (grasp, grip, pinch and gross motion), which mainly evaluates the ability of the affected hand to handle objects of different sizes, weights and shapes. ARAT requires a standardized assessment toolbox, consisting of 19 items with a full score of 57, and each item is scored in a 4-point order [0: unable to complete any part of the task within 60 s,1: complete part of the task within 60 s, 2: The task is completed, but the difficulty is very high or the time is too long (5 ~60 s),3 points: the normal completion within 5 s. Each of ARAT's subscales is arranged in a hierarchical order, testing the most difficult items first, then the easiest, and then increasing the items in turn. The higher the score, the better the feature.”

Reviewer 2 Report

First of all, I would like to congratulate the authors of the manuscript for their work. The design and usefulness of this evaluation system may be of great interest to the scientific and rehabilitation community. However, there are several modifications that would be advisable to increase the quality and comprehensibility of the manuscript.

MATERIAL AND METHODS:

Taking into consideration the subsequent analyses in the results section, it might be convenient to include a flow chart explaining and detailing the sample sizes of the different phases.

In addition, given the number of results from the multiple analyses they have performed, I believe it is appropriate to make a diagram that visually explains the evaluation stages.

I consider it necessary to include in the methods section the specific bibliographic references of at least the following evaluation tests: FMA-EU: FMA; Brunnstrom Scale and ARAT.

In the statistical analysis section, there is no reference to the procedure for calculating the sample size of the study, which it would be advisable to include.

In this same section, it is considered appropriate to calculate the effect size of the statistical differences identified.

RESULTS

Section 3.1 indicates that a total of 24 patients participated. This distribution of the number of participants has not been previously reflected in the study methodology. What was the procedure for assigning this sample size?

In Table 3, the sample size of the groups evaluated with the different tests is different. It is necessary to justify in the methodology section, as well as in the discussion, these variations in the number of participants for each analysis.

As in the previous comment, Table 4 includes different sample sizes of participants for the measurement of the variables. Justifications for these variations in the number of subjects should be included.

Table 3.5 analyses the differences, but it would be advisable to include the effect size. Also, in this section, in line 240, I think they have forgotten to indicate "x" before “±s” (“…and were  described as ±s”).

In section 3.6, the analysis performed is non-parametric, so the effect size of the statistically significant differences should be indicated as in the differential analysis tables. In this table, a notes section should be included below, indicating what the calculated Z-statistic refers to.

In Table 7 you must also indicate the name of the Z statistic.

The wording of lines 276 and 278 should be revised.

DISCUSSION

On line 295, they indicate "this instrument. However, I believe that the authors are referring to more than one evaluation instrument, so they should revise the wording of this sentence to improve the comprehensibility of the text.

Given that the internal consistency analysis of the repeated measures on the movements performed with the thumb in the initial phase of the analysis did not show significant results, I consider it necessary to further develop their reflection in the discussion.

REFERENCES

In the references section, quote number 5 is repeated twice.

Author Response

Dear Editor,

Thank you for your email dated 22 November 2022 to inform us about the reviewers’ comments.

All the authors have seriously discussed all these comments. According to the

Reviewers’ comments, we have tried our best to modify our manuscript to meet the

requirements of your journal. In this revised version, changes to our manuscript within the

document were all highlighted in red colored text. Point-by-point responses are

listed below this letter. Your kind consideration is highly appreciated.

Yours sincerely,

Baolan Wang and Xiaofeng Lu

brainsci-2050099

Application of multi-dimensional intelligent visual quantitative assessment system to evaluate hand function rehabilitation in stroke patients

Response to Reviewer 2 Comments

Comments and Suggestions for Authors

First of all, I would like to congratulate the authors of the manuscript for their work. The design and usefulness of this evaluation system may be of great interest to the scientific and rehabilitation community. However, there are several modifications that would be advisable to increase the quality and comprehensibility of the manuscript.

MATERIAL AND METHODS:

Point 1: Taking into consideration the subsequent analyses in the results section, it might be convenient to include a flow chart explaining and detailing the sample sizes of the different phases. 

Response 1: We agree with the reviewer’s suggestion, we add a flow chart describing details about this project. Please see below, and also see Figure 1 in the manuscript.

Point 2: In addition, given the number of results from the multiple analyses they have performed, I believe it is appropriate to make a diagram that visually explains the evaluation stages.

Response 2: Thank you for the suggestion, we visualized all the results in the study. Please see Figures 5-10 in the manuscript.

Supplementary Table 1. Consistency of the MDIVQAS for repeated measures to assess hand function.

Movement (n = 24)

First measurement

Second measurement

Third measurement

F

p

Wrist ulnar deviation

23.00 (4.75 to 32.75)

21.50 (6.25 to 32.75)

19.50 (4.75 ~ 30.75)

0.326

0.711

Wrist dorsiextension

35.50 (21.25 to 43.50)

36.50 (15.75 to 43.50)

32.50 (20.50 ~ 44.75)

2.489

0.098

Finger adduction and abduction

113.00 (58.00 to 146.75)

98.00 (62.25 ~ 149.25)

119.00 (43.75 ~ 139.25)

0.430

0.643

Forearm pronation

82.50 (25.50 to 88.00)

83.00 (27.00 ~ 86.75)

79.00 (22.00 to 87.00)

0.976

0.374

Forearm supination

81.00 (13.00 to 88.00)

79.00 (10.50 to 88.00)

78.00 (14.50 to 88.00)

1.666

0.204

Cylindrical grip

13.00 (4.50 to 19.75)

12.50 (4.75 to 19.00)

14.00 (5.25 to 21.00)

2.858

0.071

Spherical grip

36.00 (7.00 to 69.50)

39.00 (9.25 ~ 64.75)

38.50 (5.00 to 65.75)

0.214

0.753

Thumb abduction

15.00 (4.50 to 23.75)

16.50 (3.75 to 22.75)

17.00 (5.50 to 23.00)

3.348

0.120

Thumb flexion and extension

44.00 (12.25 to 65.00)

43.50 (9.25 ~ 62.75)

41.00 (11.50 ~ 63.75)

0.066

0.925

Thumb rotation

28.00 (10.00 to 39.75)

24.00 (7.75 ~ 38.00)

27.50 (10.00 to 37.25)

3.603

0.045*

Figure 5. Consistency of the MDIVQAS for repeated measures to assess hand function. A. Wrist ulnar deviation; B. Wrist dorsiextension; C. Finger adduction and abduction; D. Forearm pronation; E. Forearm supination; F. Cylindrical grip; G.  Spherical grip; H. Thumb abduction; I. Thumb flexion and extension. J. Thumb rotation

Supplementary Table 2. Correlation between MDIVQAS and FMA-W/H, Brunnstrom (hand), and ARAT

Evaluation methods

FMA-UE

(n = 82)

FMA-W/H

(n = 82)

Brunnstrom (hand)

(n = 72)

ARAT

(n = 36)

Multi-dimensional intelligent visual quantitative assessment system

Correlation coefficient (R)

0.690**

0.796**

0.895**

0.747**

p

0.000

0.000

0.000

0.000

Note:*p < 0.05,**p < 0.01.

Figure 6. Correlation between MDIVQAS, FMA-W/H, Brunnstrom (hand) and ARAT. A. Correlation between MDIVQAS and FMA-UE; B. Correlation between MDIVQAS and FMA-W/H; c. Correlation between MDIVQAS and Brunnstrom (hand); D. Correlation between MDIVQAS and ARAT.

Supplementary Table 3. Correlations between MDIVQAS and protractor measurement.

Movement

r

p

Forearm pronation (n=81)

0.974**

0.000

Forearm supination (n=81)

0.973**

0.000

Ulnar deviation of wrist (n=81)

0.763**

0.000

wrist dorsiextension (n=81)

0.790**

0.000

Finger adduction and abduction1 (n=37)

0.832**

0.000

Finger adduction and abduction2 (n=37)

0.916**

0.000

Finger adduction and abduction3 (n=37)

0.748**

0.000

Finger adduction and abduction4 (n=37)

0.815**

0.000

Sum of finger adduction and abduction (n=37)

0.908**

0.000

Note: *p < 0.05,**p < 0.01; finger adduction abduction 1: angle between the thumb and index finger of the affected hand, finger adduction and abduction 2: angle between the index finger and the middle finger of the affected hand, finger adduction and abduction 3: angle between the middle finger and the ring finger of the affected hand, finger adduction and abduction 4: angle between the ring finger and the little finger of the affected hand.

Figure 7. Correlation between MDIVQAS and protractor measurement.  A. Forearm pronation; B. Forearm supination; C. Ulnar deviation of wrist; D. wrist dorsiextension; E. Finger adduction and abduction1; F. Finger adduction and abduction2; G. Finger adduction and abduction3; H. Finger adduction and abduction4; I. Sum of finger adduction and abduction

Supplementary Table 4. Comparison of differences between FMA-UE, FMA-W/H, Brunnstrom and ARAT before and after treatment. 

Before the treatment

(n = 46)

After treatment

(n = 46)

t

p values

95%CI

FMA-UE

39.67±15.977

48.44±17.118

-4.233**

0.003

-13.559~-3.996

FMA-W/H

14.00±6.500

17.78±7.965

-3.640**

0.007

-6.171~-1.384

Brunnstrom (hand)

3.07±1.597

3.87±1.771

-7.284**

0.000

-1.027~-7.284

Brunnstrom (upper limb)

3.46±1.361

4.24±1.369

-7.622**

0.000

-0.989~-0.576

ARAT

40.00±17.692

44.00±18.486

-2.502*

0.037

-7.686~-0.314

Note:*p < 0.05,**p < 0.01.

Figure 8. Comparison of differences of FMA-UE, FMA-W/H, Brunnstrom and ARAT be-fore and after treatment. A. FMA-UE; B. FMA-W/H; C. Brunnstrom (hand); D. Brunnstrom (upper limb); ARAT.

Supplementary Table 5. Comparison the differences of the AROM in finger adduction and abduction. With protractor before and after rehabilitation treatment

Movement

Before treatment

(n = 37)

After treatment

(n = 37)

Z

p values

Finger adduction and abduction 1

10 (0 to 40)

25 (0 to 48)

3.157**

0.002

Finger adduction and abduction 2

5 (0 to 20)

15 (0 to 23)

2.905**

0.004

Finger adduction and abduction 3

5 (0 to 12.5)

6 (0 to 17)

2.766**

0.006

Finger adduction and abduction 4

2 (0 to 20)

15 (0 to 22.5)

2.747**

0.006

Sum of finger adduction and abduction

32 (0 to 152)

100 (0 to 166)

3.971**

0.000

 Figure 9. Comparison the differences of the AROM in finger adduction and abduction. With protractor before and after rehabilitation treatment. A. Finger adduction and abduction 1; B. Finger adduction and abduction 2; C. Finger adduction and abduction 3; D. Finger adduction and abduction 4; E. Sum of finger adduction and abduction. Z represents the effect size of the two-sample K-S test (see methods).

Supplementary Table 6. Comparison of differences before and after rehabilitation treatment using MDIVQAS

action

Before the treatment

(n = 47)

After treatment

(n = 47)

Z

p values

Wrist ulnar deviation

48.39 (14.78-65.58)

46.15 (10.34-70.27)

2.621**

0.008

Wrist dorsiextension

47.21 (14.46 to 74.52)

61.29 (20.41-84.62)

2.961**

0.002

Forearm pronation

91.48 (41.19 to 100.00)

86.36 (40.91 ~ 100.00)

2.233*

0.024

Forearm supination

80.11 (13.75 to 100.00)

88.75 (8.11 ~ 100.00)

3.393**

0.001

Cylindrical grip

46.59 (16.19 to 74.32)

58.33 (23.53-83.87)

3.852**

0.000

Spherical grip

45.36 (11.42 to 75.55)

69.40 (14.73-87.73)

2.839**

0.004

Thumb abduction

40.67 (20.63 to 68.34)

58.97 (8.33-82.14)

2.177*

0.029

Thumb flexion and extension

95.09 (32.42 to 146.20)

100.52 (12.55 ~ 159.70)

2.788**

0.005

Thumb rotation

54.25 (15.94 to 84.22)

66.33 (15.93-85.54)

2.904**

0.004

Finger adduction and abduction 1

31.82 (6.85 to 69.99)

45.64 (12.44 ~ 91.48)

3.077**

0.002

Finger adduction and abduction 2

25.00 (9.55 to 62.50)

43.99 (13.84 ~ 83.75)

2.178*

0.029

Finger adduction and abduction 3

25.93 (7.28 to 51.19)

50.84 (8.08 to 69.11)

3.115**

0.002

Finger adduction and abduction 4

32.50 (6.07 to 56.02)

47.81 (12.26 ~ 68.61)

2.510*

0.012

Sum of finger adduction and abduction

195.47 (37.55 to 329.37)

286.10 (53.59 ~ 400.46)

3.554**

0.000

Figure 10. Comparison of differences before and after rehabilitation treatment using MDIVQAS. A. Wrist ulnar deviation; B. Wrist dorsiextension; C. Forearm pronation; D. Forearm supination; E. Cylindrical grip; F. Spherical grip; G. Thumb abduction; H. Thumb flexion and extension; I. Thumb rotation; G. Finger adduction and abduc-tion 1; K. Finger adduction and abduc-tion 2; L. Finger adduction and abduc-tion 3; M. Finger adduction and abduc-tion 4; N. Sum of finger adduction and abduction. Z represents the effect size of the two-sample K-S test (see methods).

Point 3: I consider it necessary to include in the methods section the specific bibliographic references of at least the following evaluation tests: FMA-EU: FMA; Brunnstrom Scale and ARAT.

Response 3: Thank you for the suggestion, we add the specific bibliographic references of FMA-EU, FMA, Brunnstrom Scale and ARAT at the methods part. Please see below, and also see lines 170-194 in the manuscript.

2.2

“② Measuring AROM with a protractor: A universal protractor was used to measure the forearm pronation, forearm supination, ulnar deviation, wrist dorsiextension and the angle between the fingers of the five fingers[9].

③FMA-UE [15, 16]: It mainly includes movement, speed, coordination and reflex activities, with a total of 66 points, and each item is scored on a 3-level scale, that is, 0 points, unable to perform; 1 point, partially implemented; 2 points, fully implemented. Among them FMA-W/H is a part of the FMA rating scale, which evaluates the wrist and hand. There are 12 items in total, each item is 0~2 points, full score is 24 points. The higher the score, the better the motor function of the upper limb is indicated.

④Brunnstrom Scale [7, 17]: upper limb and hand parts, each part is divided into stage I-VI, the higher the level, the better the motor function. Stage Ⅰ: no exercise; Stage Ⅱ: slight flexion; Stage Ⅲ: Flexion but not extension; Stage Ⅳ: The thumb can be pinched and loosened, and the fingers can be extended semi-randomly in a small area;Ⅴ: Can do spherical or cylindrical grip, can be free to extend the whole finger, but the range of size is not equal; Stage Ⅵ: Full range extension of various grips, but with less speed and accuracy than the healthy side.

⑤ARAT [18, 19]: Consisting of 4 subscales (grasp, grip, pinch and gross motion), which mainly evaluates the ability of the affected hand to handle objects of different sizes, weights and shapes. ARAT requires a standardized assessment toolbox, consisting of 19 items with a full score of 57, and each item is scored in a 4-point order [0: unable to complete any part of the task within 60 s,1: complete part of the task within 60 s, 2: The task is completed, but the difficulty is very high or the time is too long (5 ~60 s),3 points: the normal completion within 5 s. Each of ARAT's subscales is arranged in a hierarchical order, testing the most difficult items first, then the easiest, and then increasing the items in turn. The higher the score, the better the feature.”

Point 4: In the statistical analysis section, there is no reference to the procedure for calculating the sample size of the study, which it would be advisable to include. In this same section, it is considered appropriate to calculate the effect size of the statistical differences identified.   

Response 4: Thank you for the suggestion. In line 269 of the manuscript, we add the R package that we used to calculate the sample size “The pwr package in R was used to analyze the required sample size in the study”. In each section of the Results, we add information of sample size calculating, please see below, also see lines 274-278, 301-305, 346-350.

“3.1. Reliability of MDIVQAS

The consistency test preset large effect size f=0.4[21], statistical testing power 1-β=0.8, significance level a=0.05, and at least 18 subjects were required. Considering the possibility of sample loss in the process of clinical research, Therefore, on this basis, the sample size was appropriately increased by 10% [22], and the results showed that at least 20 subjects were needed. This sample size was used to guide the content consistency test of this study.

3.2.1 Correlation between MDIVQAS, FMA-W/H, Brunnstrom and ARAT assessment

The correlation was statistically large effect size f2=0.35 [21], statistical test power 1-β=0.8, significance level a=0.05, at least 30 subjects are needed, considering the possibility of sample loss in the process of clinical study, the sample size is appropriately increased by 10% on this basis [22], the results show that at least 33 subjects are needed. This sample size was used to guide the correlation test of this study.

3.3.1 Comparison of differences of MDIVQAS, FMA-UE, FMA-W/H, Brunnstrom and ARAT before and after treatment

The difference test between the two groups was presupposed to have a large effect size d=0.8 [21], statistical testing power 1-β=0.8, and significance level a=0.05, and at least 26 subjects were required. Considering the possibility of sample loss in the process of clinical research, the sample size was appropriately increased by 10% [22]. The re-sults showed that at least 29 participants were required.”

RESULTS

Point 5:Section 3.1 indicates that a total of 24 patients participated. This distribution of the number of participants has not been previously reflected in the study methodology. What was the procedure for assigning this sample size?

In Table 3, the sample size of the groups evaluated with the different tests is different. It is necessary to justify in the methodology section, as well as in the discussion, these variations in the number of participants for each analysis.

As in the previous comment, Table 4 includes different sample sizes of participants for the measurement of the variables. Justifications for these variations in the number of subjects should be included.

Response 5: Thank you for pointing that out. Among the 82 enrolled patients, 24 were randomly selected for the consistency test of repeated tests according to the required sample size according to the statistical calculation. The flow chart of the sample size is shown in Figure 1. Please see below for specific changes. We also justified the number of subjects in the first paragraph of each corresponding result. Please see below, also see lines 274-278, 301-318, and 346-359.

Point 6: Table 3.5 analyses the differences, but it would be advisable to include the effect size. Also, in this section, in line 240, I think they have forgotten to indicate "x" before “±s” (“…and were  described as ±s”).

Response 6:Thank you for pointing that out. We have added x before ±s in line 354 of the manuscript. We visualized all the results, where Figures 6-8 represent Tables 3-5, respectively.

 Figure 6. Correlation between MDIVQAS, FMA-W/H, Brunnstrom (hand) and ARAT. A. Correlation between MDIVQAS and FMA-UE; B. Correlation between MDIVQAS and FMA-W/H; c. Correlation between MDIVQAS and Brunnstrom (hand); D. Correlation between MDIVQAS and ARAT.

Figure 7. Correlation between MDIVQAS and protractor measurement.  A. Forearm pronation; B. Forearm supination; C. Ulnar deviation of wrist; D. wrist dorsiextension; E. Finger adduction and abduction1; F. Finger adduction and abduction2; G. Finger adduction and abduction3; H. Finger adduction and abduction4; I. Sum of finger adduction and abduction

Figure 8. Comparison of differences of FMA-UE, FMA-W/H, Brunnstrom and ARAT be-fore and after treatment. A. FMA-UE; B. FMA-W/H; C. Brunnstrom (hand); D. Brunnstrom (upper limb); ARAT.

Point 7:In section 3.6, the analysis performed is non-parametric, so the effect size of the statistically significant differences should be indicated as in the differential analysis tables. In this table, a notes section should be included below, indicating what the calculated Z-statistic refers to.

Response 7: Thank you for the suggestion. Since the data in Table 6 does not conform to the normal distribution, the comparison between the two groups uses Wilcoxon's Kolmogorov-Smirnov Z test. We add a note in line 377 of the manuscript: “Z represents the effect size of the two-sample K-S test (see methods)”

Point 8: In Table 7 you must also indicate the name of the Z statistic.

Response 8: Thank you for the suggestion. Since the data in Table 6 does not conform to the normal distribution, the comparison between the two groups uses Wilcoxon's Kolmogorov-Smirnov Z test. We add a note in line 398 of the manuscript: “Z represents the effect size of the two-sample K-S test (see methods)”

Point 9: The wording of lines 276 and 278 should be revised.

Response 9: Thank you for pointing that out. We changed “It is suggested that the multi-dimensional intelligent visual quantitative assessment system could sensitively assess the subtle and clinically meaningful changes in the patient's hand function.” to “It is suggested that MDIVQA could sensitively assess changes in patients’ hand function.”. In line 388-389 of the manuscript.

DISCUSSION

Point 10: On line 295, they indicate "this instrument. However, I believe that the authors are referring to more than one evaluation instrument, so they should revise the wording of this sentence to improve the comprehensibility of the text.

Response 10: Thank you for your advice, I agree with the reviewer. We re-written the discussion part, please see below, also in lines 435-471 of the manuscript.

“Hand function plays an important role in People's Daily life, affecting people's working, eating, dressing, modifying and other activities. The improvement of hand and upper limb function will maximize the recovery of overall function and improve the quality of life of stroke patients. Effective rehabilitation requires objective, quantitative, effective and reliable rehabilitation assessment [23]. Photoelectric capture technology in intelligent evaluation tool is considered as the gold standard of human motion analysis [24]. MDIVQAS in this study is a newly developed intelligent evaluation method using optical intelligent capture technology. This system is a hand function assessment tool jointly developed by Huashan Hospital Affiliated to Fudan University and Shanghai University. It uses optical intelligent motion capture equipment and computer vision technology to conduct hand modeling and hand evaluation, and obtain three-dimensional spatial data and motion vector information of each point of fingers, palms and wrists. At present, it has been able to carry out specific intelligent analysis algorithms for the 10 movements of five fingers adduction and abduction, wrist ulnar deviation, wrist dorsiextension, spherical grip, cylindrical grip, thumb flex-ion and extension, thumb abduction, thumb rotation, forearm pronation and forearm supination, and the exercise parameters of the healthy hand angle value, the affected hand angle value, and the affected hand/healthy hand ratio were analyzed. At present, the feasibility study and quantitative evaluation application of the equipment with small samples of normal volunteers have been carried out [26]. In the early stage, the research team tested the semi-reliability and duplicate reliability in terms of reliability, and the reliability coefficients are both >0.850, indicating that the system has good re-liability, consistency and stability.  The reliability of the 10 actions of MDIVQAS showed a statistically significant difference in the reliability of the evaluators (P<0.01), indicating that the reliability of MDIVQAS retest was high, and good and stable results could be obtained by repeating the measurement in a short period of time. In terms of validity test, the content validity test of MDIVQAS in the early stage of our research team showed that all 10 movements were common hand dysfunction after stroke, the I-CVI of the entry level was 1, and each action had a high correlation with the total score (P<0.01), suggesting that it had certain evaluation value. The structural validity test adopts exploratory factor analysis, a total of 1 common factor is extracted, and the cumulative variance contribution rate is > 60% according to the functional component of the action, indicating that the system has good structural validity and can well reflect the hand motor function, and the structural validity test results show that MDIVQAS is single-dimensional in terms of evaluation content, has strong pertinence, and is suitable for quantitative evaluation of hand function. In terms of the convergence validity test, the AVE of the system > 0.500, indicating that it has good convergence validity [27].”

Point11: Given that the internal consistency analysis of the repeated measures on the movements performed with the thumb in the initial phase of the analysis did not show significant results, I consider it necessary to further develop their reflection in the discussion.

Response 11: We agree with the reviewer’s suggestion. We added this section to the discussion, list as below, also in lines 472-490 of the manuscript.

“The team expanded the sample size of the previous study, and at the same time used MDIVQAS to test the intra-group consistency of 10 movements of the affected hand of stroke patients, and Cronbach's Alpha > 0.9, indicating good internal consistency. The repeatability of 10 actions was measured, and the results showed that the differences in the repeated measurement of 9 actions were not statistically significant (all P > 0.05), indicating that the system had good repeatability. Only one of the movements (thumb rotation) had a statistically significant difference (F=3.603, P=0.045), indicating that the repeatability of this action needs to be explored. In conclusion, the above evidence shows that MDIVQAS has good confidence in the assessment of hand function after stroke. The consideration of the results of thumb rotation may be related to the fact that the current development of computer vision and pattern recognition algorithms has not reached the level of good recognition of any complex actions. When the dysfunctional hand of stroke patients is the hand, and the Brunnstrom stage ≥IV on this side, the thumb rotation action is more flexible than the healthy hand (the hand that builds the model), the data exceeds the modeling range and the data accuracy is reduced. According to the experience of the evaluator, when the hand function is relatively good or recovers to a certain extent, due to the flexibility of the hand, the completion of the action is better than the healthy side, and the accuracy of the data is reduced. It is recommended to debug and rectify the evaluation and measurement methods of the above actions.”

Reviewer 3 Report

This article has a novel content but needs a number of corrections to be made by the authors before publication.

This publication does not define a clear objective. "This study focused on the application of multi-dimensional intelligent visual quantitative assessment  system in the rehabilitation assessment of hand function in stroke patients" instead the development of the same shows a validation process of the instrument.  The objective should be adapted to the development of the work. The conclusion of the study should also be modified according to the objective and content of the article.

In the introduction, the presentation of demographic data focuses only on the Chinese population. In line 36 it is stated: "China is the largest developing country, accounting for about one fifth of the world's population, and the number of current stroke patients ranks first in the world[8]" this sentence should be deleted as it does not provide relevant information that has not been previously pointed out.  Also data on the incidence and impact of stroke in the rest world's population should be presented.   

When describing the assessment tools, some very relevant ones are omitted, used in the most recent studies that should be included or at least cite why they are discarded. 

In line 50 it is stated: "However, the above evaluation methods are affected by subjective factors, difficult to identify accurately, and lack of quantification". This statement is not supported by any scientific study, a bibliographic reference should be included to support this statement, and the authors are dismissing the tools that are then used to justify the consistency of MDIVQAS.

The material and methods section includes data from the final sample that should not be in this section, these data should be presented in the results section, line “total of 82 patients, including 57 males (69.5%) and 25 females 67 (30.5%), completed the evaluation of the multi-dimensional intelligent visual quantitative 68 assessment system with average age of (54.29±13.12) years”. On the other hand, the method of selection of the sample to be included is not indicated, as well as the calculation of the sample size.

The test administration procedure needs to be detailed. Excessive technical characteristics of the MDIVQAS are described; it would be correct to summarize them and indicate the test administration procedure.

In the discussion section it is indicated that MDIVQAS measures functionality when what it indicates is the movement of the forearm, wrist and some hand movements. Not functionality of the hand, since neither opposition of the thumb, nor gripper, nor intrinsic hand movements such as rotation and translation are assessed. The manual dominance of the subject is not considered and no data on muscle balance is provided. All these concepts should be dealt with in the discussion section. This section should be better defined, according to the results, what benefits the use of MDIVQAS provides and remove comparisons with other therapeutic robots, line 296 "Other researchers have designed a wearable hand function rehabilitation robot that can qualitatively assist patients to complete the exercise training tasks of four-finger stretching and thumb abduction[17]".

The conclusion should be rephrased in relation to the objective and correspond to the actual results found, indicating specific movements detected by the image analysis system with statistical significance and which require further study. The authors should eliminate the generalities asserted about functionality.

Author Response

Dear Editor,

Thank you for your email dated 22 November 2022 to inform us about the reviewers’ comments.

All the authors have seriously discussed all these comments. According to the

Reviewers’ comments, we have tried our best to modify our manuscript to meet the

requirements of your journal. In this revised version, changes to our manuscript within the

document were all highlighted in red colored text. Point-by-point responses are

listed below this letter. Your kind consideration is highly appreciated.

Yours sincerely,

Baolan Wang and Xiaofeng Lu

brainsci-2050099

Application of multi-dimensional intelligent visual quantitative assessment system to evaluate hand function rehabilitation in stroke patients

Response to Reviewer 3 Comments

Point 1: This article has novel content but needs a number of corrections to be made by the authors before publication.

Response 1: Thanks to the reviewers for their suggestions, we have considered and revised the wording of the full text.

Point 2: This publication does not define a clear objective. "This study focused on the application of multi-dimensional intelligent visual quantitative assessment  system in the rehabilitation assessment of hand function in stroke patients" instead the development of the same shows a validation process of the instrument.  The objective should be adapted to the development of the work. The conclusion of the study should also be modified according to the objective and content of the article.

Response 2:Thanks to the reviewer’s advice. We redefined the objective in the abstract part. please see lines 12-15 and lines 25-26 in the manuscript.

We also clarified our objective in introduction section, list as below, also see lines 59-67 in the manuscript.

“MDIVAQAS is the core technology of hand motion calculation jointly developed by Huashan Hospital affiliated to Fudan University and Shanghai University with completely independent intellectual property rights. Through the optical smart cap-ture and the integration of computer vision technology, complete animation action standard hand more guidance, the contralateral national health model and subject to lateral hand evaluation process, belongs to the field initiative, and it can complete the quantitative assessment of hand function. The purpose of this study was to verify the reliability of MDIVQAS, and to observe the evaluation effect of the system in clinical application compared with traditional evaluation methods.”  

Point 3: In the introduction, the presentation of demographic data focuses only on the Chinese population. In line 36 it is stated: "China is the largest developing country, accounting for about one fifth of the world's population, and the number of current stroke patients ranks first in the world[8]" this sentence should be deleted as it does not provide relevant information that has not been previously pointed out.  Also data on the incidence and impact of stroke in the rest world's population should be presented.   

Response 3: Thanks to the reviewer’s suggestion. We re-written the first paragraph. Please see below. Also in lines 30-37 of the manuscript.

“Stroke is a major non-communicable disease that harms the health of people all over the world. As of 2019, stroke is the second leading cause of death worldwide and the third leading cause of death caused by disability. From 1990 to 2019, the disability rate caused by stroke increased by 32.0% [1] .70% to 90% of patients after stroke have upper limb dysfunction [2, 3] and 60-70% of them are hand dysfunction [1][4]. As one of the most important organs of the human body, the hand is involved in many daily activities, and its function accounts for 90% of the upper limb function. Hand dysfunction seriously affects patients' daily life and social participation ability[5].”

Point 4: When describing the assessment tools, some very relevant ones are omitted, used in the most recent studies that should be included or at least cite why they are discarded. 

Response4:  Thanks to the reviewer’s pointing that out. We included recent studies in the introduction. Please see below, also find in lines 39-53 of the manuscript.

“At present, the commonly used methods for assessing hand function after stroke are mainly qualitative or semi-quantitative, which mainly include: Protractor measurement of Active Range of Motion (AROM), FMA-UE, FMA-W/H, Brunnstrom, ARAT and so on. The above evaluation methods have been widely used in clinical practice for a long time, and their reliability and validity are also widely recognized. However, the actual application is easily influenced by personal experience[6-9]. Therefore, there is great interest in developing an automated system to achieve intelligent, objective and quantitative assessment of hand function rehabilitation after stroke. In the last two decades, significant advances have been made in human motion measurement and analysis, providing the technical basis for automated assessment of upper limb and hand function [10]. In recent years, modern methods for quantitative hand function assessment include 3D motion capture systems, Kinect technology, rehabilitation robots and micro sensors, etc. However, the above instruments are still lacking in large sample studies, and some instruments are too bulky and have not been widely used in clinical practice[11-13].”

Point 5: In line 50 it is stated: "However, the above evaluation methods are affected by subjective factors, difficult to identify accurately, and lack of quantification". This statement is not supported by any scientific study, a bibliographic reference should be included to support this statement, and the authors are dismissing the tools that are then used to justify the consistency of MDIVQAS.

Response 5: We agree with the reviewer’s suggestion. We re-written the introduction part. Please see below. Also, in lines 37-58 of the manuscript.

“Rehabilitation therapy is an important method to restore hand function, and accurate treatment is based on accurate assessment. At present, the commonly used methods for assessing hand function after stroke are mainly qualitative or semi-quantitative, which mainly include: Protractor measurement of Active Range of Motion (AROM), FMA-UE, FMA-W/H, Brunnstrom, ARAT and so on. The above evaluation methods have been widely used in clinical practice for a long time, and their reliability and validity are also widely recognized. However, the actual application is easily influenced by personal experience[6-9]. Therefore, there is great interest in developing an automated system to achieve intelligent, objective and quantitative assessment of hand function rehabilitation after stroke. In the last two decades, significant advances have been made in human motion measurement and analysis, providing the technical basis for automated assessment of upper limb and hand function [10]. In recent years, modern methods for quantitative hand function assessment include 3D motion capture system, Kinect technology, rehabilitation robots and micro sensors, etc. However, the above instruments are still lacking in large sample studies, and some instruments are too bulky and have not been widely used in clinical practice[11-13].

In clinical rehabilitation work, intelligent, quantitative, accurate and efficient hand function assessment tools are needed to provide a better choice for clinical practice. Therefore, this study uses a newly developed intelligent assessment method, the ap-plication of multi-dimensional intelligent visual quantitative evaluation system (MDIVQAS) to objectively and quantitatively assess the function of the affected hand after stroke.”

Point 6: The material and methods section includes data from the final sample that should not be in this section, these data should be presented in the results section, line “total of 82 patients, including 57 males (69.5%) and 25 females 67 (30.5%), completed the evaluation of the multi-dimensional intelligent visual quantitative 68 assessment system with average age of (54.29±13.12) years”. On the other hand, the method of selection of the sample to be included is not indicated, as well as the calculation of the sample size.

Response 6: Thank you for the advice.

(1)We moved “a total of 82 patients, including 57 males (69.5%) and 25 females 67 (30.5%), completed the evaluation of the multi-dimensional intelligent visual quantitative 68 assessment system with the average age of (54.29±13.12) years” to the result part. Please find in lines 274-280

(2)The selection of the sample size is shown in Flowchart 1, and the calculation of the specific sample size is as follows:

“3.1. Reliability of MDIVQAS

The consistency test preset large effect size f=0.4[21], statistical testing power 1-β=0.8, significance level a=0.05, and at least 18 subjects were required. Considering the possibility of sample loss in the process of clinical research, Therefore, on this basis, the sample size was appropriately increased by 10% [22], and the results showed that at least 20 subjects were needed. This sample size was used to guide the content consistency test of this study.” In this study, the sample size in Table 1 is 24 cases to meet the most basic sample size requirements.

3.2.1 Correlation between MDIVQAS, FMA-W/H, Brunnstrom and ARAT assessment

The correlation was statistically large effect size f2=0.35 [21], statistical test power 1-β=0.8, significance level a=0.05, at least 30 subjects are needed, considering the possi-bility of sample loss in the process of clinical study, the sample size is appropriately in-creased by 10% on this basis [22], the results show that at least 33 subjects are needed. This sample size was used to guide the correlation test of this study.

3.3.1 Comparison of differences of MDIVQAS, FMA-UE, FMA-W/H, Brunnstrom and ARAT before and after treatment

The difference test between the two groups was presupposed to have a large effect size d=0.8 [21], statistical testing power 1-β=0.8, and significance level a=0.05, and at least 26 subjects were required. Considering the possibility of sample loss in the process of clinical research, the sample size was appropriately increased by 10% [22]. The results showed that at least 29 participants were required.”

 Figure 1. Flow chart of each stage in the study

Point 7: The test administration procedure needs to be detailed. Excessive technical characteristics of the MDIVQAS are described; it would be correct to summarize them and indicate the test administration procedure.

Response 7: Thank you for the suggestion. We listed the details below, also in lines 114-169 of the manuscript.

“The specific assessment methods as follows:

①MDIVQAS:Based on the pathological motor characteristics of hemiplegic hand and a set of post-stroke hand function rehabilitation evaluation actions corresponding to Brunnstrom scale, Fugl-Meyer Rating Scale and range of motion measurement, it is a computer-aided technology-based assessment tool. Using the comprehensive quantitative evaluation method of healthy hand modeling and comparison evaluation of the affected hand, the 3D spatial position and motion vector information of various joints of the phalanx, metacarpal and wrist were acquired in real time with the help of video equipment, and then various motion parameters of the hand joint were analyzed as the system parameters of the hand function evaluation standard. In order to prevent the ambiguity and subjectivity in the guidance process of the standard movement demonstration, At the bottom left of the screen, there is a 3D animation of the action being evaluated to achieve a consistent demonstration of standard hand movements. The assessment items included three parts as forearm, wrist and hand, with a total of 10 movements, including ulnar wrist deviation, wrist dorsiextension, five fingers adduction and abduction, forearm pronation, forearm supination, spherical grip, cylindrical grip, thumb flexion and extension, thumb abduction and thumb rotation.

When the patient was seated, the patient information was first input, such as number, name, age, brief medical history, assessment results of common scales, etc. The pathological information of the patient included healthy hand, affected hand, stroke type, stroke brain area, etc. The hospital information includes the name, address, and contact information of the hospital. Then, the movements to be evaluated were selected. The patient placed the healthy hand and the affected hand in the evaluation device at the same time, and the same visual and optical acquisition devices were configured for the healthy hand and the affected hand respectively. In the first step of evaluation, the patient's healthy hand was guided by the standard animated hand to complete the extraction of the motion characteristics and node parameters of the patient’s healthy hand and the modeling of the healthy hand model. The 3D animated hand part was captured by Maya software. The standard hand video obtained the foreground hand through the background learning algorithm, and the spatial motion trajectory was tracked by the hand particle filter. In the second step of assessment, the affected hand completed the extraction of motion characteristics and joint parameters in the affected hand working area. In the third step of evaluation, the multi-dimensional hand motion parameters of the patient's affected hand to be evaluated were comprehensively analyzed, and the joint motion of the affected hand, the percentage of joint motion of the affected hand in the healthy hand, and the evaluation time were automatically calculated. In the third step of evaluation, the multi-dimensional hand movement parameters of the patient's hand to be evaluated are comprehensively analyzed, and the joint range of motion of the affected hand, the percentage of joint mobility of the affected hand to the healthy hand, and the time of evaluation are automatically calculated. Evaluate twice, and the system will automatically select the better angle result for saving. MDIVQAS flow block diagram, see Figure 2.

Figure 2. workflow diagram

Point 8: In the discussion section it is indicated that MDIVQAS measures functionality when what it indicates is the movement of the forearm, wrist and some hand movements. Not functionality of the hand, since neither opposition of the thumb, nor gripper, nor intrinsic hand movements such as rotation and translation are assessed. The manual dominance of the subject is not considered and no data on muscle balance is provided. All these concepts should be dealt with in the discussion section. This section should be better defined, according to the results, what benefits the use of MDIVQAS provides and remove comparisons with other therapeutic robots, line 296 "Other researchers have designed a wearable hand function rehabilitation robot that can qualitatively assist patients to complete the exercise training tasks of four-finger stretching and thumb abduction[17]".

Response8: Thank you for the suggestion. We have made changes in the Discussion section. Please see below, also in lines 455-470 of the manuscript.

“Photoelectric capture technology in intelligent evaluation tools is considered as the gold standard of human motion analysis [24]. MDIVQAS in this study is a newly de-veloped intelligent evaluation method using optical intelligent capture technology. This system is a hand function assessment tool jointly developed by Huashan Hospital Affiliated to Fudan University and Shanghai University. It uses optical intelligent mo-tion capture equipment and computer vision technology to conduct hand modeling and hand evaluation, and obtain three-dimensional spatial data and motion vector in-formation of each point of fingers, palms and wrists. At present, it has been able to car-ry out specific intelligent analysis algorithm for the 10 movements of five fingers ad-duction and abduction, wrist ulnar deviation, wrist dorsiextension, spherical grip, cy-lindrical grip, thumb flexion and extension, thumb abduction, thumb rotation, forearm pronation and forearm supination, and the exercise parameters of the healthy hand angle value, the affected hand angle value, and the affected hand/healthy hand ratio were analyzed. At present, the feasibility study and quantitative evaluation applica-tion of the equipment with small samples of normal volunteers have been carried out [26].”

Point 9: The conclusion should be rephrased in relation to the objective and correspond to the actual results found, indicating specific movements detected by the image analysis system with statistical significance and which require further study. The authors should eliminate the generalities asserted about functionality.

Response 9: Thank you for the suggestion. We restated the results and conclusions according to the objective. Please see below, also see lines 288-292, 321-325, 409-412, 554-556 of the manuscript.

“3.1. Reliability of MDIVQAS

The 24 patients enrolled in the group completed 10 movements using MDIVQAS, and each movement was repeated 3 times by the same examiner. Cronbach's Alpha coefficient method was used to analyze the internal consistency of the assessment sys-tem. All the 10 actions were greater than 0.9, indicating that the internal homogeneity reliability of MDIVQAS was excellent and the internal consistency was good.

3.2.1 Correlation between MDIVQAS, FMA-W/H, Brunnstrom and ARAT assessment

Statistical analysis was performed on the correlation between the percentage of the affected hand on the unaffected side measured by MDIVQAS and the various scales. Bivariate Correlations, Pearson Correlations and two-tailed test were used for statistical parameters. The results showed that MDIVQAS was strongly correlated with FMA-W/H, Brunnstrom (hand) and ARAT (r > 0.5, p < 0.01).

3.3.2 Comparison of the difference between MDIVQAS and protractor measurement of AROM in the increase of joint motion before and after treatment

The reactivity of the two evaluation methods to the treatment effect was basically the same, and there was no statistical significance in the increase of the range of motion between the two evaluation methods before and after treatment (P > 0.05), indicating that the two evaluation methods had the same reactivity to the treatment effect.

In conclusion, MDIVQAS has good reliability and validity in the evaluation of hand function in stroke, as well as good evaluation of treatment effect. However, there are some shortcomings in the application that need to be further studied and improved.”

Round 2

Reviewer 2 Report

The authors have made an excellent revision of the comments made in my first review. The quality of the manuscript has improved substantially, especially in the methodological and results sections. 

I consider it appropriate to congratulate the research team for the work done and to indicate that the article meets the quality required for publication.